# Fault Displacement Hazard Analysis Based on Probabilistic Seismic Hazard Analysis for Specific Nuclear Sites

**Tamás János Katona** [1,*] **, László Tóth** [2] **and Erzsébet Győri** [3]

1. Faculty of Engineering and Information Technology, University of Pécs, 7624 Pécs, Hungary
2. GEORISK Earthquake Engineering Ltd., 1118 Budapest, Hungary; toth@georisk.hu
3. Kövesligethy Radó Seismological Observatory, Institute of Earth Physics and Space Science, 1112 Budapest, Hungary; gyori@seismology.hu
* Correspondence: katona.tamas@mik.pte.hu

**Featured Application: Methodology has been developed and applied with the aim to contribute to the justification of safety of the Paks nuclear site in Hungary.**

**Abstract:** Permanent ground displacements/deformations caused by earthquakes can seriously challenge the safety of the nuclear power plants. The state-of-the-art hazard analysis methods provide a fault displacement hazard curve, i.e., the annual probability of given measure of displacement will be exceeded. The evaluation of ground displacement hazard requires great effort, empirical evidence, and sufficient data for the characterization of the fault activity and capability to cause permanent surface displacement. There are practical cases when the fault at the site area revealed to be active, and, despite this, there are no sufficient data for the evaluation of permanent ground displacements hazard and for judging on the safety significance of permanent ground displacement. For these cases, a methodology is proposed that is based on the seismotectonic modelling and results of the probabilistic seismic hazard analysis. The method provides conservative assessment of the annual probability of fault displacement that allows the decision whether permanent displacement hazard is relevant to nuclear power plant safety. The feasibility and applicability of the method is demonstrated for the Paks site, Hungary.

**Keywords:** permanent ground displacement; probabilistic seismic hazard assessment; probabilistic fault displacement hazard assessment; on-fault displacement; distributed faulting; nuclear safety

## 1. Introduction

Fault displacement hazard has been a crucial siting issue from the beginning of the construction of commercial nuclear power plants. According to older regulatory approach the site should be abandoned, if a fault near the site judged for capable to cause permanent surface displacement. In the recent siting practice, the hazard should be evaluated and should be accounted for in the design basis, and the safety should be proven (see, International Atomic Energy Agency requirements and guidance SSR-1 [1] and SSG-9 [2] and U.S. Nuclear Regulatory Commission 10 CFR 100 [3]).

The attributes for capability of the faults to generate surface displacement are as follows [2,3]:

(1) Evidence of significant past movement or movements of a recurring nature within such a period that it is reasonable to conclude that further movements at or near the surface may occur. This time interval is fixed by the regulations. Generally, it varies between 10,000 to 100,000 years (see, e.g., [3]). Note, the significance should be qualified considering the effect surface movement on the integrity and function of safety related structures of nuclear power plant, i.e., the significance is matter of nuclear engineering rather than geoscience.

(2)   Structural relationship exists with a known capable fault such that movement of the one fault may cause movement of the other at the surface.

(3)   If the maximum potential magnitude of the earthquake, associated with a seismogenic faults at the site vicinity are sufficiently large to cause surface movement.

The procedure for assessing the tectonic surface fault rupture hazard for nuclear facilities is given by the ANSI/ANS-2.30-2015 standard [4,5]. A comprehensive guidance for the justification of plant safety with respect to permanent ground displacement (further indicated as PGD) has been published by the Japan Nuclear Safety Institute [6].

A procedure for screening out the PGD hazard was presented by Gürpinar et al., 2017 [7] that is based on the maximum potential magnitude of the fault considered. This concept has also been discussed by Katona [8]. According to this, for the screening, the probability of non-zero surface rupture $P(sr \neq 0 | M_u)$ should be calculated as is as proposed by Wells and Coppersmith [9]. The hazard can be neglected, if the annual probability of non-zero surface rupture is less than $10^{-7}$, that corresponds to 1% of acceptable annual probability for core damage or to the acceptable limit for severe accidents for new reactors. If the PGD hazard cannot be screened out with high confidence, the annual probability of exceedance should be calculated $\lambda(D \geq D_0)$ for the direct movement $D$, and the $\lambda(d \geq d_0)$ for the distributed fault movement $d$. There are two conceptually different approaches for the PGD hazard analysis: the displacement based and the earthquake-based approaches [5,6]. For both approaches detailed characterization of fault activity and potential movements on the fault is needed.

There are practical cases, where pre-Quaternary faults mapped at the site vicinity revealed to be re-activated in the Quaternary period. However, there are no manifestations of permanent surface movement or the observed PGD is negligible small. Moreover, the available data for characterization of the fault capability to cause surface rupture are insufficient. Difficult to interpret cases have been reported, for example, in [10] for the Bohemian Massive and very recently for the Paks site in Hungary [11]. The practical motivation of this paper was to contribute to the clarification of permanent fault displacement hazard issue for the Paks site in Hungary.

The Paks site is in the middle of the Pannonian Basin (46.34 N, 18.51 E). The site was selected for the Paks Nuclear Power Plant (Paks NNPP) more than sixty years ago. The historically credible maximum intensity was defined as MSK-64 grade 6. In the early nineties, a probabilistic seismic hazard assessment was made based on the full-scope geological, geophysical, seismological, and geotechnical investigations of the site. Since 1995, the site seismic hazard has been reviewed and updated every ten years in the frame of periodic safety reviews and the stress-test in 2011. Although the site is a licensed nuclear site, for the new Paks 2 plant to be constructed at the site, a full-scope site investigation and hazard evaluation have been implemented recently applying state-of-the-art techniques and methodologies. Presentation of the results of this extensive and complex investigations would exceed the frame of this paper. A brief summary is given below highlighting the scientific and practical motivation of the research.

The Paks site is in the mid-part of Pannonian basin that is a region with typical diffuse seismicity (see, e.g., in [12]). As a result of the investigations for Paks nuclear site, it was known that the Kapos fault zone in central Hungary crosses the site vicinity area. Therefore, the fundamental question of former and recent site seismotectonic investigations was, whether some segments of the known fault zone at the site vicinity have been re-activated during Quaternary period. Recent paleoseismic investigations revealed that the eastern segment of the Kapos zone, the so called Dunaszentgyörgy–Harta fault zone shows neotectonic activity (Horváth et al. [11]). This is a broad, sinistral strike-slip shear zone consisting of predominantly NE–SW and ENE–WSW striking individual faults. The strike-slip kinematics of this shear zone is clearly indicated by the observed internal "flower structure" of the individual fault zones, as well as by the associated secondary fault pattern (Riedel faults).

For the illustration of the geology and tectonic features of the near regional area (~25 km in radius [2] of the site), the schematic tectonostratigraphic profile of the area and the definition of geological horizons (1–6) mapped in the 3D geological model Figure 1.

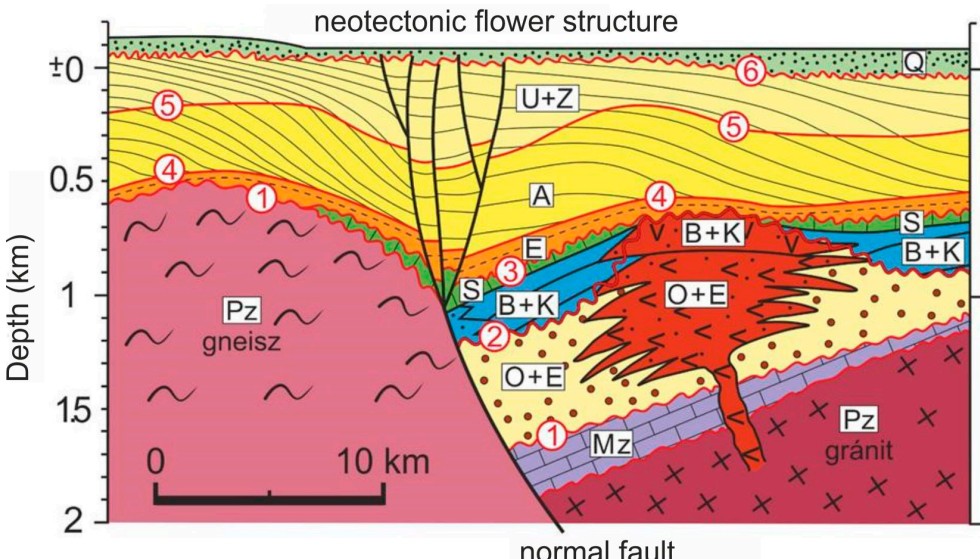

**Figure 1.** Schematic tectonostratigraphic profile of the area and the definition of geological horizons (1–6) mapped in the 3D geological model. The horizons surveyed are the following boundary surfaces and unconformities: (1) top of basement, (2) M1 clastic Miocene top, (3) M2 marine Miocene top, (4) Endrőd top, (5) Algyő top, (6) Quaternary base. Q = Quaternary formations; U + Z = "Upper Pannonian" Újfalu és Zagyva Form; A = "Lower Pannonian" Algyő Form; E = "Lower Pannonian Endrődi Form. consisting of deep basinal marls; S = Sarmatian formations; B + K = marine coastaland open marine Badenian és Karpatian formations; O + E = Lower-Miocene Ottnangian and Eggenburgian siliciclastic; K + B = Karpatian Tar Dacite Tuff and Badenian Mátra Andesite Form; O + E = Ottnangian(?) and Eggenburgian Gyulakeszi Rhyolite Tuff and Mecsek(/Paks) Andesite Form.; Mz/Pz = Mesozoic and Palaeozoic basement rocks.

The young geological deformations (after Wórum et al., [13]) and recent earthquakes (1995–2020) in the area 100 km in radius around the Paks NPP site are shown in Figure 2. On the other hand, there are no historical and instrumental earthquake records of any magnitude in the site vicinity, which is defined in line with [2] as a 5 km in radius scale. No event recorded within 10 km; 13 earthquakes within 25 km; 87 earthquakes within 50 km and 2484 known earthquakes within 100 km.

Based on the 3D geological-structural model high-resolution 2D and pseudo-3D shallow high-resolution geophysical surveys were performed to locate the possible indications of Quaternary re-activation and near surface movement. The subsequent trenching proved a near surface movement with ~10 mm vertical and ≤25 mm horizontal displacement with age about twenty thousand years. There are no historical or instrumental records of earthquakes in the site vicinity area. Moreover, the microseismic monitoring does not indicate activity during already three decades of monitoring and the GPS geodesy does not show significant tectonic movement (≤0.1 mm/year) [2]). Considering the complex set of information that includes geological, geophysical, seismological, and paleo-seismological aspects and the mapped tectonic movement, a qualitative conclusion was made by Horvath et al. in [11] and in the site safety study [14] that the fault could not cause significant for nuclear safety surface movement. Here, the significance should be understood in sense of nuclear safety (see the safety guide SSG-9 [2]).

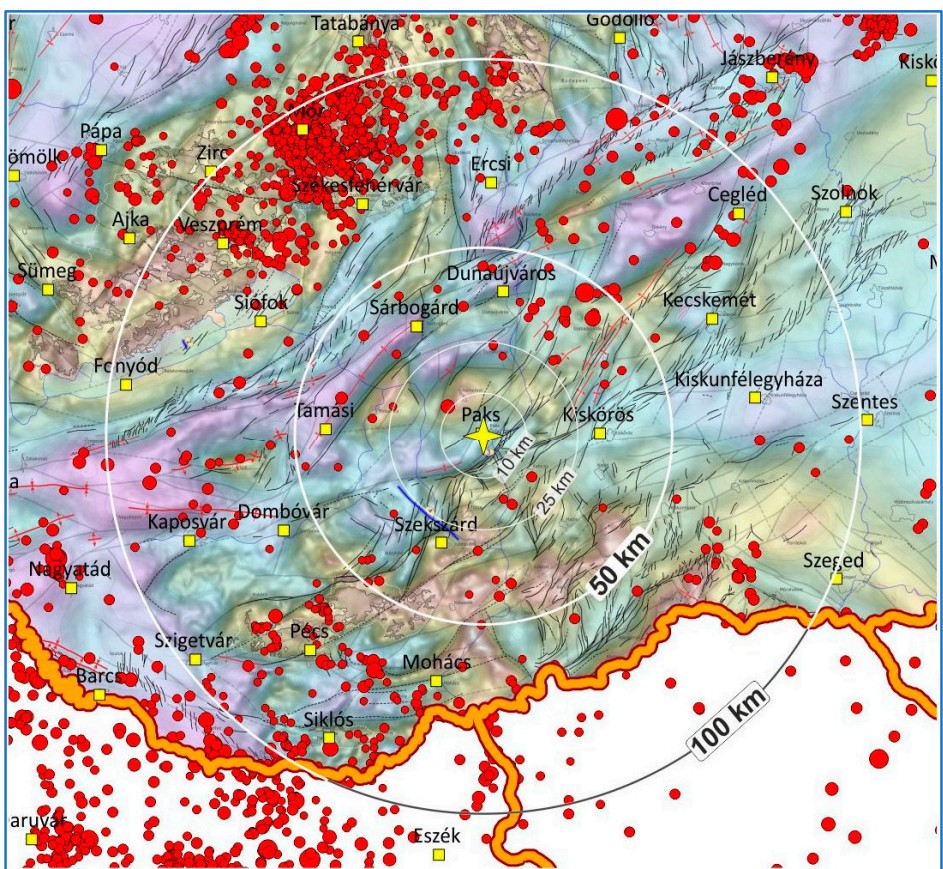

**Figure 2.** Young geological deformations (after Wórum et al., 2020 [13]) and recent earthquakes (1995–2020) in the vicinity of the Paks NPP site.

Notwithstanding the conclusion above, in accordance with international requirements (see [1,2]), a quantitative evaluation of PGD hazard should be performed, and the PGD hazard curve should be provided for the assessment of nuclear safety significance for the new built plan and for the operation of the Paks Nuclear Power Plant and the spent fuel interim storage facility located within the site area (one square kilometer [2]).

Understanding the practical need and considering the objective limitation of data on fault activity, a methodology for probabilistic fault displacement hazard analysis has been developed in the paper that is based on the seismotectonic modelling and results of probabilistic seismic hazard analysis (PSHA). The proposed methodology considers the uncertainties related to fault characterization and applicable for conservative estimation of the permanent ground displacement hazard curve for both on-fault and distributed contribution to the PGD hazard. The analysis has a theoretical importance since this would be a first of a kind study for permanent fault displacement in the Pannonian basin.

## 2. The Methodology for Probabilistic Fault Displacement Hazard Assessment Based on Probabilistic Seismic Hazard Assessment

### 2.1. The Concept and Basis of Development of the Methodology

If there are not sufficient data for the adequate characterization of the activity of suspicious fault or faults, the evaluation of permanent surface displacement hazard can be performed based on the magnitude-distance disaggregation of seismic hazard. Thus, the logic-tree modeling of seismogenic sources and the results of the probabilistic seismic hazard evaluation could be the basis of PGD hazard analysis, instead of the logic-tree modelling of fault activity as it is required by the probabilistic fault displacement hazard analysis. Therefore, the concept for the evaluation of the fault displacement hazard for the specific conditions of the study site is based on three existing and widely accepted methods:

- Probabilistic seismic hazard assessment method (see, Ordaz and Salgado-Gálvez R-CRISIS v20 [15]). It should be noted, a state-of-the-art PSHA exists for the site. The first PSHA has been performed for the site in 1995 [16] and regularly updated every ten years in the frame of periodic safety reviews of the Paks (see, e.g., in [17]) and post-Fukushima stress-test and very recently for the site licence of the new plant (see in [14]).
- Earthquake based probabilistic fault displacement hazard evaluation method, see [4] and Youngs et al. [5] since this method is based on the characterisation of seismogenic sources/faults similar to the PSHA.
- Strike-slip displacement evaluation method of Petersen et al. [18] (see also, Chen and Petersen [19] and Thio and Somerville [20]).

The methods above and their applications are published in several papers. Therefore, the methods are considered as well known.

### 2.2. Theoretical Background for the Evaluation of the Principal Fault Contribution to PGD Hazard

For strike-slip faults Petersen et al. [18] proposed for the rate $\nu(D \geq D_0)_{xyz}$ at which the displacement, D, on the fault exceeds a specified amount, $D_0$, the following equation:

$$\nu(D \geq D_0)_{xyz} = \alpha(m) \int_{m,s} f_{M,S}(m,s) P[sr \neq 0|m] \times \int_r P[D \neq 0|z,r,sr \neq 0] \times P[D \geq D_0|l/L,m,D \neq 0] f_R(r) dr dm ds \quad (1)$$

by Chen and Petersen [19].

Here, the $\nu(D \geq D_0)_{xyz}$, annual rate of exceedance $D \geq D_0$; $x, y$ are the coordinates of the site area and, $z$ denotes the dimension of the site. The $L$ is the rupture length, and $l$ is the length on the fault where the closest distance from the site $r$ is measured. The $\alpha(m)$ is the rate of earthquake with magnitude $m$. The $f_{M,S}(m,s)$ is the probability density function of the earthquake with magnitude $m$ that is due to rupture with distance $s$ from the end of the fault. The $P[sr \neq 0|m]$ is the probability of non-zero surface movement happens due to earthquake with magnitude $m$ that can be calculated as proposed by Wells and Coppersmith [9]. The $P[D \neq 0|z,sr \neq 0]$ is the probability for the non-zero displacement will be at the site $z$ under the non-zero rupture condition. The $P[D \geq D_0|l/L,m,D \neq 0]$ is the conditional probability of $D \geq D_0$ if the earthquake of magnitude, $m$ happens with location $l/L$. The $f_R(r)$ is the distribution density of the possible distance of the site area to the fault. The term $P[D \neq 0|z,sr \neq 0]$ is the conditional probability of non-zero surface movement at the site of area $z$ and that is at a distance r from the fault.

Interpretation of the geometrical variables in the Equation (1) is shown in Figure 3.

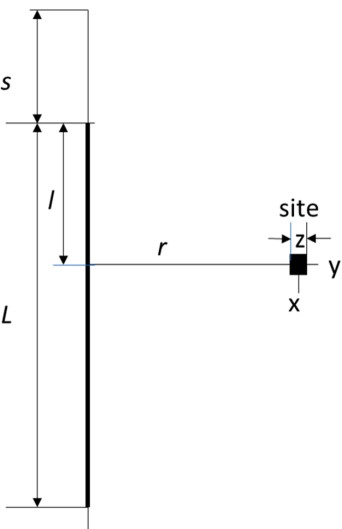

**Figure 3.** Definition of geometrical variables (Petersen et al., 2011 [18]).

For strike-slip faults Petersen et al. [18] developed bilinear, quadratic, and elliptical models for magnitude-distance relationship applicable for principal-fault displacements in the form:

$$\ln(D) = b \cdot x^* + a \cdot m + c \tag{2}$$

Here, the $x^* = \sqrt{1 - \frac{1}{0.5^2}\left[\frac{l}{L} - 0.5^2\right]^2}$ depends on the location of the site with respect to the fault as it is seen in Figure 3. If the site is located at the centreline to $L$, the $l/L = 0.5$ and $x^* = 1$. If the site located on the line perpendicular to the endpoint of the fault surface projection $l/L = 0.0$ and $x^* = 0$. The coefficients are $b = 3.3041$, $a = 1.7927$ and $c = -11.2192$, with a standard deviation on $\ln(D)$ of 1.1348.

The distributions and conditional probabilities needed for Equation (1) cannot be developed for the fault zone at the study site because of lack of data as indicated in the introduction. Even the rate $\alpha(m)$ cannot be assessed, since a single, twenty thousand years old indication of Quaternary re-activation of the fault has only be identified.

Further conservative simplifications should be introduced into the Equation (1).

It can be assumed, that the site is located at the centerline to the fault trace, i.e., $l/L = 0.5$ and the fault is ruptured at full length, i.e., $s = 0$, see, Figure 3. Consequently, the product of conditional probabilities $P[D \neq 0|z, r, sr \neq 0] \times P[D \geq D_0|l/L, m, D \neq 0]$ will reduce to $P(D \geq D_0|r, m)$. Further, it is assumed that the bivariate distribution density of magnitudes and distances, $p(r, m)$ is known for the site. Based on these assumptions, for the annual rate of exceedance, $\nu(D \geq D_0)$ the following equation can be written:

$$\nu(D \geq D_0) = \int_{m_0}^{m_u} \alpha(m) P(sr \neq 0|m) \int_0^{r_u} P(D \geq D_0|r, m) p(r|m) dr dm. \tag{3}$$

Here, $\alpha(m)$ is the annual rate of earthquake with magnitude $m$ for the principal fault considered. $P(D \geq D_0|r, m)$ is the conditional probability of the $D \geq D_0$ due to the earthquake of magnitude $m$, at the distance to the site, $r$. The $P(sr \neq 0|m_j)$ is the conditional probability of non-zero surface rupture. The $m_0$ is the minimum magnitude to be accounted for in the PGD calculation, $m_u$ is the maximum possible magnitude associated to the fault, $r_u$ is the maximum distance to the site that is reasonable to consider in the calculation for the principal fault contribution. It should be noted, that the Equation (3) is a continuous representation of the equation published by Thio and Somerville [20] $\nu(D \geq D_0) = \sum_{j=0}^{M} \alpha(m_j)\left[\sum_{k=1}^{N} P(D \geq D_0|r_k, m_j) P(sr \neq 0|m_j) P(r_k|m_j)\right]$, where the index $j = 0$ corresponds to lower $j = M$ to the maximum possible magnitude $m_u$ and $m_0$, respectively.

### 2.3. The Proposed Methodology for On-Fault Displacement Evaluation

The study site is located within the of mid-Pannonian area with typical diffuse seismicity. Although the pre-Quaternary faults are mapped, the data are insufficient for the parametrization of the recent activity of the presumably reactivated in the Quaternary fault in the site area. The probabilistic seismic hazard analysis exists for the site is based on comprehensive seismotectonic modelling of source areas (see [14]). Therefore, the methodology for PGD hazard evaluation developed below is based on probabilistic seismic hazard analysis available for the site. This method is a specific realization of "earthquake-based approach" [5,6] for probabilistic permanent displacement hazard analysis

As part of the PSHA, the seismic hazard can be disaggregated into magnitude-distance bins for every selected hazard level $\lambda$ [1/a] (see [15]). The probability weight associated to magnitude $Mw$ and distance $R_{JB}$ pairs characterize the contribution of the magnitude distance pairs/bins to the hazard.

The disaggregation of hazard at given level $\lambda$ can be transformed into a bivariate probability density function $\wp(Mw, R_{JB})$. In discrete case, the joint probability mass function is given for magnitude and distance bins $(Mw_i \div R_{JB_j})$ with n $\times$ m size weights

matrix, $[W_{i,j}]$. The marginal densities $p_m(Mw)$ and $p_m(R_{JB})$ are represented by $(w_i; Mw_i)$ and $\left(w_j; R_{JB_j}\right)$, while:

$$\sum_{i=0}^{i=i_u} [W_{i,j}] = [w_i] \text{ and } \sum_i w_i = 1 \tag{4}$$

where the upper index $i_u$ corresponds to the maximum magnitude up-to those the disaggregation is performed. The marginal density of the distances, $r$ can be calculated similarly to the Equation (4).

Further, the relation between distance between the fault trace and the site, $r$ and $R_{JB}$ should be established. The $R_{JB}$ is the Joyner-Boore distance (see [15]), the closest distance to the projection of the fault plane on the surface, see Figure 4. As it is shown in Figure 4, the $R_{JB}$ can be assessed as a conservative estimation for $r$ since it corresponds to the situation when $r$ is minimum.

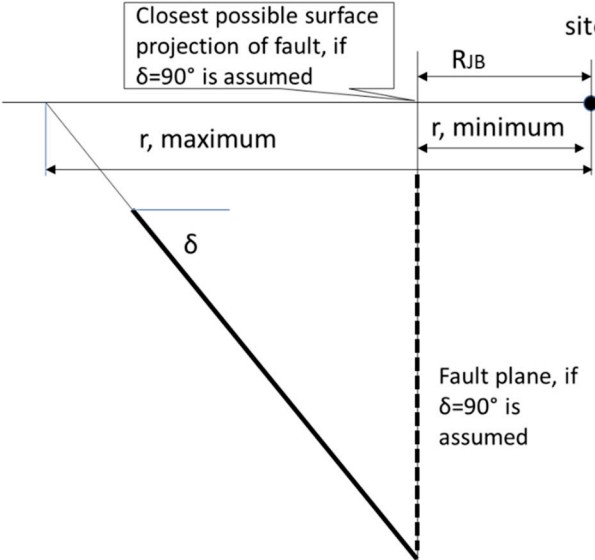

**Figure 4.** Definition of geometrical variables ($r$ is distance from the site to the mapped fault trace on the surface and $R_{JB}$ the Joyner-Boore distance).

Further, for the sake of simplicity, the variables $Mw_i$ will be replaced by $m_i$ and the $R_{JB_j}$ by $r_j$ (or by $m$ and $r$ in continuous form of equations).

For the hazard level $\lambda$, the Equation (3) can be modified as follows:

$$P(D \geq D_0)|_\lambda = \left[ \int_{m_0}^{m_u} P(sr \neq 0|m) \int_0^{r_u} P(D \geq D_0|r,m) \times p_\lambda(m,r) dr dm \right]\Bigg|_\lambda. \tag{5}$$

Integrating over $r$, the Equation (5) takes the following form:

$$P(D \geq D_0)|_\lambda = \left[ \int_{m_0}^{m_u} P(sr \neq 0|m) P(D \geq D_0|m) p_{m,\lambda}(m) dm \right]\Bigg|_\lambda. \tag{6}$$

Here the $p_\lambda(m,r)$ is the bivariate density and $p_{m,\lambda}(m)$ is the marginal distribution density of magnitudes at hazard level $\lambda$.

Equations (5) and (6) give the probability of $D \geq D_0$ at annual level $\lambda$. Note, that the disaggregation of seismic hazard at hazard level $\lambda$ is nothing else as the discrete representation of the bivariate density $p_\lambda(m,r)$ of magnitude and distances. Thus, Equations (5) and (6) can be written in discrete form as follows:

$$P(D \geq D_0)|_\lambda = \left[ \sum_{i=0}^{i=i_u} \sum_{j=0}^{j=j_u} P(sr \neq 0|m_i) P(D \geq D_0|r_j, m_j) W_{i,j} \right]\Bigg|_\lambda. \tag{7}$$

$$P(D \geq D_0)|_\lambda = \left[ \sum_{i=0}^{I=i_u} P(sr \neq 0|m_i) P(D \geq D_0|m_i) w_i \right]\Bigg|_\lambda. \tag{8}$$

Here and below the upper indexes $i = i_u$ and $j = j_u$ corresponds to the maximum magnitude and distance up-to those the disaggregation is performed.

Applying Equation (8) the expected value of the on-fault displacement $\overline{D}|_\lambda$ can be calculated for the worst-case since all earthquakes at the fault are centered to the site ($l/L = 0.5$ and $x^* = 1$) and all earthquakes are accounted for in the PGD evaluation independent from the distance from the site.

Equation (2) establishes the correlation between magnitude and $D$ that can be used in the further analysis. Thus, writing Equation (2) as:

$$D = exp(b + a \cdot m + c), \tag{9}$$

and the expected value of displacement, $\overline{D}|_\lambda$ will be:

$$\overline{D}|_\lambda = \sum_{i=0}^{i=i_u} w_i \cdot P(sr \neq 0|m_i) \cdot exp(b + a \cdot m_i + c) \Bigg|_\lambda \tag{10}$$

Due to the assumptions made above, the $\overline{D}|_\lambda$ is obviously over-conservative, since the on-fault displacement is an is a proximal phenomenon. Therefore, the magnitude distribution for the proximal to the site distance bin should be accounted for in the calculation. Let's denote the weight distribution for magnitude in the closest bin to the site in the disaggregation as $(w_i\ m_{r=0,i})$. For this case, the Equation (10) can be rewritten as follows:

$$\overline{D}|_\lambda = \sum_{i=0}^{i=i_u} w_i \cdot P(sr \neq 0|\ m_{r=0,i}) \cdot exp(b + a \cdot\ m_{r=0,i} + c) \Bigg|_\lambda \tag{11}$$

Thus, considering the standard deviation for the Equation (2), for each hazard level $\lambda$, a mean, a mean minus standard deviation and a mean plus standard deviation values can be obtained. Thus, three estimations of the hazard curve can be created for $\overline{D}$. The averaging means here that the displacement is averaged for all possible magnitudes that contributed to the seismic hazard at hazard level $\lambda$.

In this procedure, the epistemic and aleatory uncertainties on the seismotectonic modeling are accounted for in the PSHA via logic-tree technique.

### 2.4. Evaluation of Distributed Fault Displacement Hazard

Petersen et al., 2011 [18] suggested the equation below equation for the calculation of annual rate of distributed fault contribution to the PGD hazard:

$$\lambda(d \geq d_0)_{xyz} = \alpha(m) \int_{m,s} f_{M,S}(m,s) P[sr \neq 0|m] \times \int_r P[d \neq 0|r, z, sr \neq 0] \times P[d \geq d_0|r, m, d \neq 0] f_R(r) drdmds. \tag{12}$$

Here the $\lambda(d \geq d_0)_{xyz}$ is the annual probability of the off-fault displacement $d \geq d_0$ at a location $(x; y)$ within area $z$. The meaning of the probabilities and marginal distribution of distances is identical to those in Equation (1). Petersen et al. [18] suggested the following equation for the direct calculation of $d$:

$$\ln(d) = 1.4016 * m - 0.1671 * \ln(r) - 6.7991 \tag{13}$$

Here, $d$ is in centimeters and $r$ in meters, and with standard deviation of 1.1193 in *ln* units. Like the considerations above, the following equation can be written for the probability of $d \geq d_0$ at the hazard level $\lambda$:

$$P(d \geq d_0)|_\lambda = \left[ \int_{m_0}^{m_u} P(sr \neq 0|m) \int_0^{r_u} P(d \geq d_0|r, m) \times p_\lambda(m,r) drdm \right]\Bigg|_\lambda. \tag{14}$$

Equation (14) can be written for the probability of $d \geq d_0$ for distributed faulting as follows:

$$P(d \geq d)|_\lambda = \left[ \sum_{i=0}^{i=i_u} \sum_{j=0}^{j=j_u} P\Big(sr \neq 0|m_j\Big) P\Big(d \geq d_0|r_j, m_i\Big) W_{i,j} \right] \Bigg|_\lambda . \qquad (15)$$

The expected value of displacement, $\overline{d}\Big|_\lambda$ for given hazard level $\lambda$ will then be:

$$\overline{d}\Big|_\lambda = \left[ \sum_{i_i=0}^{i=i_u} \sum_{j=0}^{j=j_u} P\Big(sr \neq 0|m_j\Big) exp\big(1.4016 m_i - 0.1671 \ln(r_j) - 6.7991\big) W_{i,j} \right] |_\lambda \qquad (16)$$

Here, the findings of Livio et al. [21] and Gürpinar et al. [7] regarding potential area to be accounted for while the contribution of distributed faulting is calculated. Therefore, the calculation via Equation (16) is performed for all distance bins.

## 3. The Results of PSHA Based PGD Hazard Evaluation

### 3.1. The Input for Calculation

For each hazard level, the PSHA disaggregation provides the n × m size weights matrixes, $W_{i,j}$ for $(Mw_i \div R_{JB_j})$ bins. The marginal densities $p_m(Mw)$ and $p_m(R_{JB})$ are represented by pairs $(w_i; Mw_i)$ and $\Big(w_j; R_{JB_j}\Big)$ (see Equation (4)).

The PSHA and its results are documented in [14]. Here, the most essential outputs of the disaggregation are only presented that characterize the site seismic hazard and important are for the PGD hazard evaluation via proposed methodology.

Figure 5 shows the disaggregation of seismic hazard for the study for the level $10^{-5}$ annual probability of exceedance taken from [18].

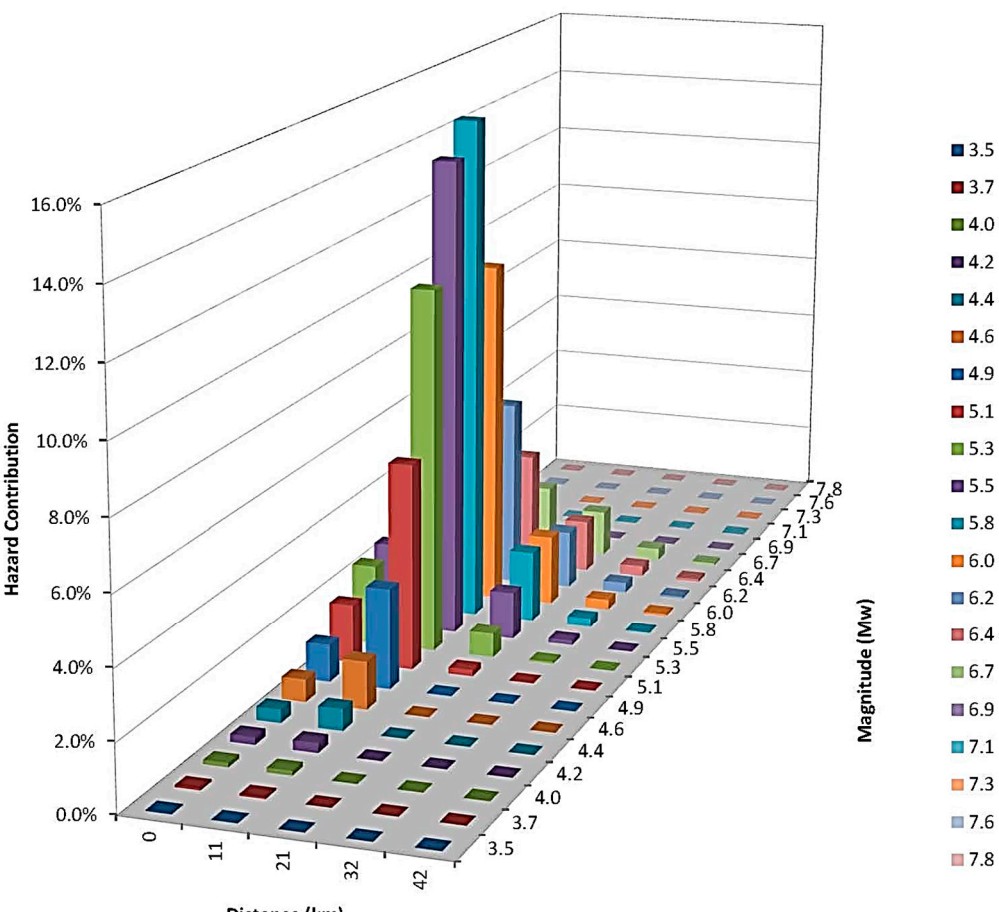

**Figure 5.** Disaggregation of the seismic hazard for $10^{-5}/a$ hazard level (color code refers to magnitudes).

The weight distribution of contributing magnitudes and $R_{JB}$ distances to the hazard levels $10^{-4}$/a to $10^{-7}$/a are shown in Figure 6a,b. Note, the weight of the proximal to the site disaggregation bin is about 10% for all hazard levels considered.

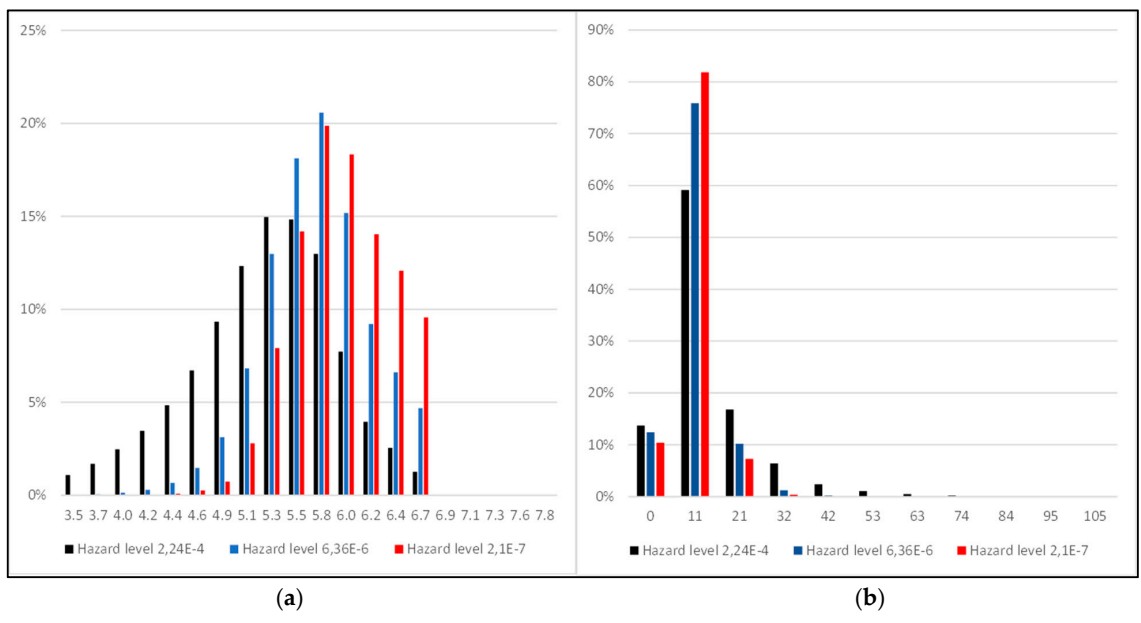

(**a**) (**b**)

**Figure 6.** Contribution of different magnitudes (**a**) and distances (**b**) to different levels of seismic hazard.

*3.2. Calculation of Hazard Curves for $\overline{D}$*

For the study site the hazard curves for $\overline{D}$ has been calculated via Equation (10). The curves are shown in Figure 7a. The hazard curve for the $\overline{D}$ calculated for the proximal disaggregation bin via Equation (11) is shown in Figure 7b.

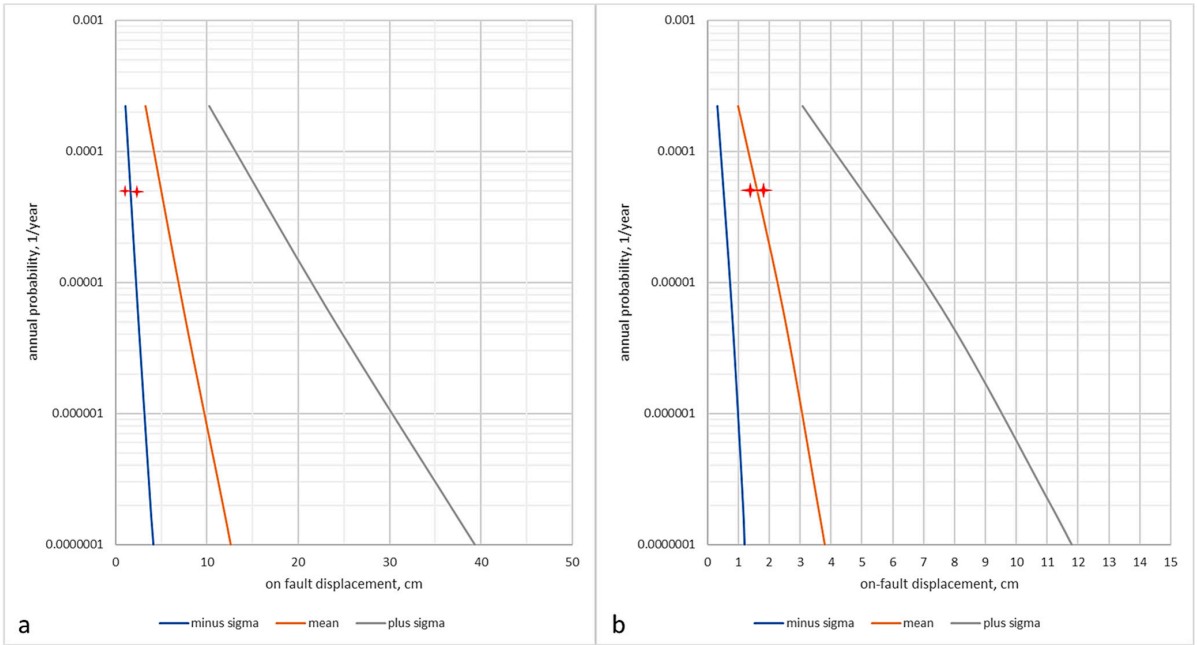

**Figure 7.** The hazard curve for on fault displacement $\overline{D}$. averaged over marginal density of magnitudes (**a**) and calculated for the proximal disaggregation bin (**b**). The observed movements found in trenching are indicated in figures by asterisk.

The curves denoted as mean and plus or minus standard deviation are the result of accounting the ε = ±1.1348 on ln(D) in Equation (2). The uncertainties of seismotectonic modelling are accounted for in the PSHA.

### 3.3. Calculation of Mean Hazard Curve for Contribution of Distributed Faulting

The mean hazard curve for the contribution of distributed fault displacement, the $\overline{d}$ has been calculated via Equation (16). The results are plotted in Figure 8.

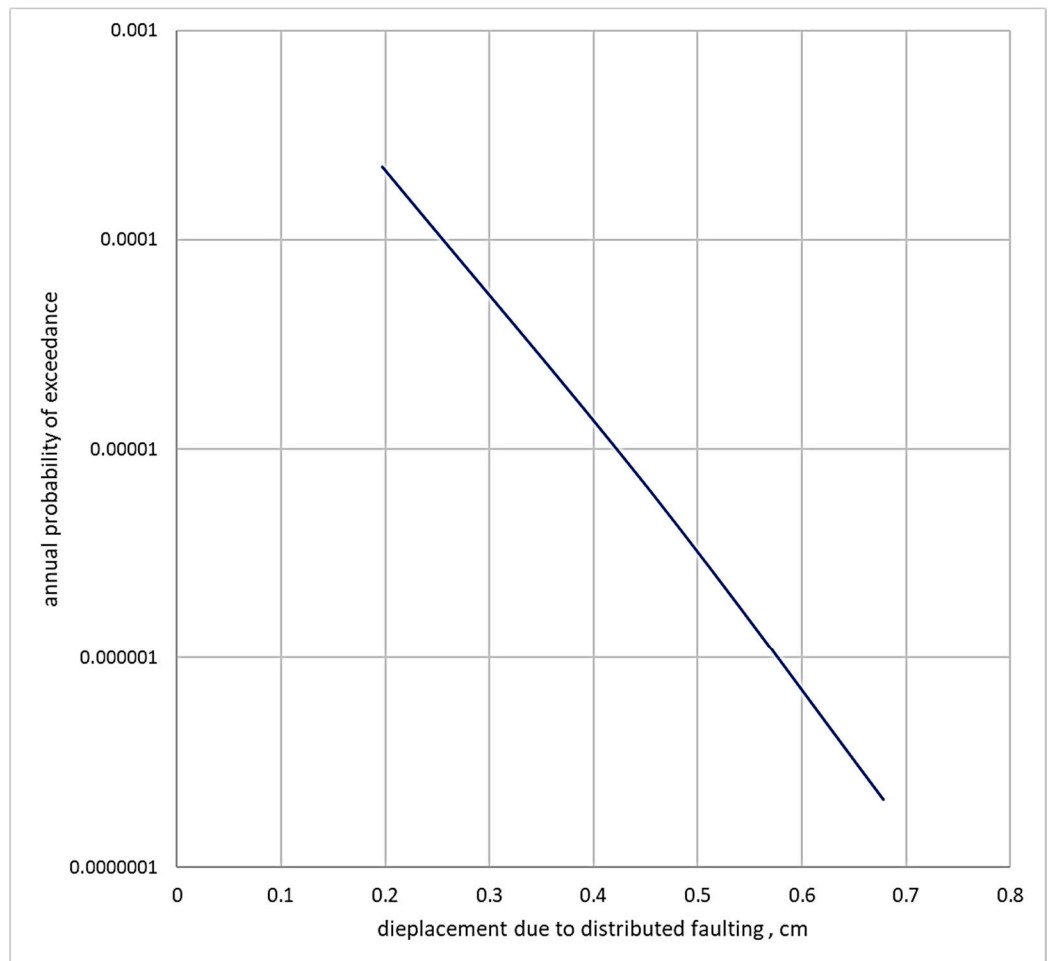

**Figure 8.** Mean hazard curve for distributed faulting displacement.

### 4. Discussion

For the study site, as it is shown in Figure 6, with decreasing hazard level, the distribution of the magnitudes gets narrower and tends to the maximum possible magnitude of the closest source. These justify the assumption to use the marginal distribution for magnitudes since the distribution of the magnitudes is getting narrow at the low hazard levels interesting from the point of view of nuclear safety.

The estimation of the exceedance probability for maximum fault displacement, $\overline{D}$ (Figure 7a) is overconservative, because all earthquakes at the fault with rupture length $L$ are assumed to be centered relative to the site ($l/L = 0.5$) and r = 0, see Figure 3. Note, the contribution of the closest bin remains around 10% for all hazard levels as it is seen in Figure 6b. Therefore, the calculation that accounts just the magnitude distribution for the proximal to the site bin ($w_i\, m_{r=0,i}$) should provide a more reasonable estimation of the displacement hazard as it is shown in Figure 7b. The conservatism of the calculation of the exceedance probability versus displacement could further be reduced if the empirical

correlation for averaged displacement $D_{ave}$ would be used in Equations (10) and (11) according to Wells and Coppersmith [22]. The above considerations on the conservativeness are valid for the estimation of the mean hazard curve for the contribution of the distributed faulting. The mean hazard curve in Figure 6 shows that the contribution of distributed faulting is much less than the on-fault displacement.

The uncertainties of the empirical correlation for $D$ are considered via standard deviations given for the corresponding empirical correlations. The epistemic and aleatory uncertainties regarding seismotectonic modelling is accounted for in the PSHA for all hazard levels. Thus, instead of the logic-tree modelling that is required by the probabilistic fault displacement hazard analysis, the epistemic uncertainties are accounted for in the logic tree modeling of the seismotectonic sources in the probabilistic seismic hazard assessment.

As it has been mentioned in the introduction, trenching reveals a late-Pleistocene (approximately 20 thousand years old) $20 \div 25$ mm horizontal movement [11]. The observed displacements are indicated in Figure 7a,b. By chance, these values sit on the curve "minus standard deviation" at $\approx 5 \times 10^{-5}$/a in case of more conservative estimation obtained by Equation (10). If the less conservative analysis is made using Equation (11), the observed indications of the movement fit to the mean hazard curve.

The coincidence of the observed in the trenching movements with the predicted one could be interpreted as proof of the adequacy of the proposed calculation procedure. Of course, these rather uncertain indications of near-surface displacement are insufficient for the empirical validation of the proposed methodology. Unfortunately, at the studied site, the only evidence of the near surface displacements is the mentioned above indications. Data or observations on surface displacements in the entire central part of the Pannonian Basin are practically missing. Therefore, there are two actions planned for systematic validation of the methodology. First, a step-by-step comparison of the proposed method with the probabilistic fault displacement hazard evaluation procedure could be performed. Validation of the methodology against the empirical data should also be a research task for the future. An option could be a trial implementation of the method for a site where both probabilistic seismic hazard assessment and probabilistic fault displacement hazard assessment have been made, as for example for the Krsko nuclear site in Slovenia, see Quittmeyer et al. [23]. However, such a comparison requiring international cooperation exceeds the frame of the recent paper.

Based on the estimated hazard curves, the significance of the PGD hazard for nuclear safety can be assessed, and an estimation could be made on the safety relevance of PGD hazard at the study site. The mean curve in Figure 7b can be used as a basis for considerations. According to the regulatory practice and international nuclear safety requirements (see, for example, the WENRA RHWG Report [24]) the annual probability $10^{-7}$ can be accepted as criterion for "practical elimination" of early/large releases for new nuclear power plants. It means, if the effects of the hazard with $10^{-7}$ annual probability could not result in the early/large releases, the hazard can be assessed as not relevant for nuclear safety. As it is seen in [6], complex analysis is needed for the evaluation of integrity of safety related structures of nuclear power plant to the PGD effects that exceeds the frames of the present work. As a thumb rule, proposed by Gürpinar et al. [7] threshold displacement can be considered. According to this, if the predicted value of PGD is about 0.1 m at $10^{-7}$/year level, the hazard would not challenge the plant safety. As it seen on the mean curve in Figure 7b, this condition is fulfilled in case of the Paks site.

## 5. Conclusions

In the paper, a procedure has been developed for the evaluation hazard curves of the on-fault displacement as well as for the contribution of distributed faulting that is based on PSHA disaggregation results. The method is applicable for any sites where the data on the activity and capability of the faults suspected to cause surface displacement are insufficient for performing a probabilistic permanent ground displacement hazard analysis as proposed by [4,5].

The proposed conservative method allows the decision whether the surface displacement should be accounted for in the design and can resolve the suspect issue of possible non-acceptable surface displacement. The proposed methodology has been applied for the Paks nuclear site in Hungary. The Late-Pleistocene displacement found by trenching fits the less conservative version of the calculated mean hazard curve both with the value and with the annual probability. According to our rough estimation using the mean hazard curve, the predicted for $10^{-7}$ annual probability displacement would not represent a challenge to the safety of nuclear power plant located at the site.

**Author Contributions:** Conceptualization, methodology, formal analysis, and original draft preparation were part of T.J.K.; data curation, the PSHA investigations, review and editing and visualization were part of L.T. and E.G. All authors have read and agreed to the published version of the manuscript.

**Funding:** This research received no external funding.

**Institutional Review Board Statement:** Not applicable.

**Informed Consent Statement:** The study did not involve humans.

**Data Availability Statement:** Data not included in the paper are available on request through contacting the corresponding author.

**Conflicts of Interest:** The authors declare no conflict of interest.

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
