# Peer review of "Fault Displacement Hazard Analysis Based on Probabilistic Seismic Hazard Analysis for Specific Nuclear Sites"

_applsci, doi:10.3390/app11157162_

Round 1

Reviewer 1 Report

This article considers a PFDHA based on PSHA methodology for a specific NPP site. This article is well developed and I could not find any comments for this article.

Reviewer 2 Report

  • Avoid informal language like ‘the early times’ – state what is meant by this with references to the early implementation of fault displacement hazard in nuclear waste storage applications
  • “Generally, it varies between 10000 to 100000 years” – provide references for this statement
  • There many grammatical errors throughout. The paper should be carefully reviewed to fix these errors.
  • “there are no historical and instrumental earthquake records in the site vicinity” – how is the ‘site vicinity’ defined – over what distances are no earthquakes recorded, and what is the approximate magnitude completeness of this statement? E.g., would a magnitude 3 at a distance of 50 km from the site count, or not?
  • “the conclusion was that the fault could not cause significant for nuclear safety surface movement” = who made this conclusion? Based on absence of seismicity?

Reviewer 3 Report

The paper presents a methodology to estimate the fault displacements, trough a probabilistic hazard analysis, in a specific site aimed to Nuclear scopes. The paper is interesting but it needs of review. I suggest authors to follow the comments in the attached PDF file  

Reviewer 4 Report

The authors present a case study for a specific nuclear site based on the evaluation of fault displacement using PSHA results.

This manuscript focuses on a relevant and timely problem, up to now investigated by means of a multidisciplinary approach.

The idea presented in this study is interesting and could represent an alternative approach to analyse many real cases. The authors apply the proposed approach to the Paks site in Hungary: in such site the difficulties to calculate the displacement hazard curves due to not sufficient data, is overcome by using the seismotectonic modelling.

In my opinion, this paper needs to be strongly improved, as concern the description of the different approaches used and combined as well as the definition of variables, the limitations due to the specific test case and so on.

The major drawback of this paper is the lack of a precise description of the investigated problem and of the strategy to solve it. Additionally, the paper is confused and difficult to read since crucial information is missing or is referred to other papers. Moreover, the references are not sufficient to frame the paper.

In the present version the paper is non eligible for publication. I recommend a strongly rewriting and resubmitting.

Specific comments:

-           A figure introducing the site, geology, the seismological condition is mandatory.

-           The captions of all figures are poor and must be improved.

-           Figure 1 should include the geometrical sketch of x*=0 case

-           Figure 2 is reporting "r" segments two times

-           Figure 3 need some additional information related to the colours scale used

-           In the main test, sometimes, the cited equation is wrong. I.e.: line 205,251, 258

-           The variable "z" represents the site dimension (a surface) so it is not necessary to use z2 or z2 (line 138, 304)

-           The index in some equations needs to be checked (Eq.7 and Eq.8)

-           the references should be formatted according to the journal requirements

-           Few typos were found in the text and should be amended.

Round 2

Reviewer 4 Report

The authors revised the manuscript follows indications and suggestions proposed.